# The Effects of Orofacial Myofunctional Therapy on Children with OSAHS’s Craniomaxillofacial Growth: A Systematic Review

**DOI:** 10.3390/children10040670

**Published:** 2023-03-31

**Authors:** Yue Liu, Jian-Rong Zhou, Shi-Qi Xie, Xia Yang, Jing-Lan Chen

**Affiliations:** College of Nursing, Chongqing Medical University, Chongqing 400016, China

**Keywords:** orofacial myofunctional therapy, oropharyngeal exercises, orofacial myofunctional reeducation, obstructive sleep apnea-hypopnea syndrome, craniofacial

## Abstract

Orofacial myofunctional therapy (OMT) is one of the therapeutic methods for neuromuscular re-education and has been considered as one of the auxiliary methods for obstructive sleep apnea hypopnea syndrome (OSAHS) and orthodontic treatment. There is a dearth of comprehensive analysis of OMT’s effects on muscle morphology and function. This systematic review examines the literature on the craniomaxillofacial effects of OMT in children with OSAHS. This systematic analysis was carried out using PRISMA (Preferred Reporting Items for Systematic Reviews and Meta-Analyses) standards, and the research was scanned using PICO principles. A total of 1776 articles were retrieved within a limited time, with 146 papers accepted for full-text perusing following preliminary inspection and 9 of those ultimately included in the qualitative analysis. Three studies were rated as having a severe bias risk, and five studies were rated as having a moderate bias risk. Improvement in craniofacial function or morphology was observed in most of the 693 children. OMT can improve the function or morphology of the craniofacial surface of children with OSAHS, and its effect becomes more significant as the duration of the intervention increases and compliance improves. In the majority of the 693 infants, improvements in craniofacial function or morphology were seen. The function or morphology of a kid’s craniofacial surface can be improved with OMT, and as the duration of the intervention lengthens and compliance rises, the impact becomes more pronounced.

## 1. Introduction

An upper respiratory tract problem that occurs while you are asleep is defined as obstructive sleep disordered breathing (SDB). Primary snoring, upper respiratory resistance syndrome, and OSAHS are all phenotypes. OSAHS is a common childhood illness that has severe health repercussions and causes young parents to worry [1]. OSAHS and sleep bruxism (SB) are two of the major phenotypes connected to sleep and oral health [2,3,4] Incidence rates for snoring and OSAHS, which climax between the ages of 2 and 6, are 27% and 5.7%, respectively [2]. The incidence of SB in children is determined to be 3.5–40.6% according to a comprehensive study of the international literature [5]. Because of the high rate of incidence, rigorous trials are needed to identify children at high risk and provide appropriate treatment [6]. Mild craniofacial deformation and hypotonia were identified as risk factors for SDB [7]. Masticatory hypotonic disorder is closely related to the severity of OSAHS and continuous snoring [8]. The relationship between sleep problems and craniofacial features is controversial [9], with most studies suggesting that breathing patterns affect the function and morphology of craniofacial muscles [10,11] and negatively affect the efficiency and how quickly the upper airway collapses while you sleep [12]. When the natural balance of the perioral muscles is disturbed, the teeth and jaws gradually adapt to the abnormal balance, resulting in abnormal oral movements and growth [13,14]. In addition, prolonged mouth breathing gradually stretches the child’s head and neck forward, increasing the anterior craniocervical angle [15,16]. The angle between the mandible and maxilla rises sharply as OSAHS severity increases, and the proportion of anterior mandibular height to total face length considerably declines [17]. Motta et al. reported craniocervical postural abnormalities in children with SB using digital photogrammetry [18]. OMTs are more comprehensive and, unlike continuous positive airway pressure (CPAP) and oral orthotics, do not just target symptom relief. By rewiring muscle activity, they also try to increase upper airway compliance and minimize open mouth breathing. OMT evolved from traditional speech therapy and uses isometric and isotonic movements to train the mouth and jaw neck in various multifunctional activities [19,20]. This results in a functional balancing of the mouth and face, the correction of poor oral habits [21], an improvement in breathing during sleep [22], a reduction in AHI by approximately 50% and 62% in adults and children [22,23], and ultimately, the promotion of growth and development [24]. There are few studies on the effects of OMT on the craniofacial region. Currently, OMT alone or in combination has achieved positive results in controlling sleep breathing symptoms [22,23,25], improving craniomaxillofacial cephalometric indices [21,26,27], enhancing muscle thickness, and boosting muscle activity [28]. For children with OSAHS who have craniofacial abnormalities, OMT may be a valuable alternative therapy. In the quickly developing specialty of dental sleep medicine, dentists may be able to assist kids with craniofacial anomalies [29]. These data cannot prove a linkage between pediatric OSAHS and craniofacial morphology due to the extremely low-to-moderate degree of confidence involved [29]. There are no systematic evaluations or meta-analyses in the effects of OMT on facial morphology in children with OSAHS. There are limited data on whether OMT can improve facial morphology in children with OSAHS. The effectiveness of OMT is debatable; because there are few studies on the field, intervention strategies differ widely, there is a high risk of bias, and the research content is of poor quality. In this study, the impacts of OMT on craniofacial function and morphology in minors with OSAHS were investigated and the characteristics and shortcomings of OMT intervention programs were discussed.

## 2. Materials and Methods

### 2.1. Protocol

The Preferred Reporting Items for Systematic Reviews and Meta-Analyses (PRISMA) statement is followed by this procedure [30]. The PICO question was “In kids with OSAHS, is OMT helpful in changing craniofacial morphology and function compared to controls or no treatment?”.

### 2.2. Eligibility Criteria

Prospective, retrospective, and non-randomized controlled studies were all acceptable research designs for evaluating craniomaxillofacial characteristics. We included papers from PubMed, Web of Science, Cochrane Library, and China National Knowledge Infrastructure up until 1 November 2022, and there was no defined article language limit. Studies that reported cases with insufficient knowledge as well as review papers, opinion pieces, and letters that failed to provide original data were removed. 

The search strategy was developed using the PICOS structure: (1) The population consists primarily of minors; (2) OMT is an intervention; (3) Comparison: patients before and after OMT or without; (4) Outcome: facial bone growth disorders, abnormal functional and morphological development of craniomaxillofacial region; (5) Study design: clinical controlled trials, prospective research, and retrospective study.

#### Primary Outcomes

(1) Cephalometric indicators [31,32]: SNA, SNB, ANB, PP-MP, SN-MP, SN-PP, SN-OP, OP-MP, FMA, N-Me, SN-Gn, SNGoGn, GoGn, ArGoMe, ArGo, N-ANS, ANS-Me, S-Go, MP-H, 1-NA, 1. NA, 1. NB, NB, SPAS, PAS, C3-H, overbite, and overjet.

(2) Muscle function assessment: the Iowa Oral Performance Instrument (IOPI) [33,34] and orofacial myofunctional evaluation with scores (OMES) [35].

(3) Sleep breathing assessment: Apnea hypopnea index (AHI) in polysomnography (PSG); OSA-18 quality-of-life questionnaire (OSA-18) [36]. 

### 2.3. Information Sources and Search Strategy

Four electronic databases were searched up to 1 November 2022 (Table 1). A brief description of our search outputs is shown in Figure 1. The full collection of articles that were included is presented in Table 2. 

### 2.4. Study Selection

Title and abstract were used to filter the search strategy’s preliminary results. The full texts of essential publications were checked for eligibility and exclusion criteria (Figure 1).

### 2.5. Data Collection Process and Data Items

Two researchers independently filled up data extraction forms with information on the type of investigator, year, study type, sample characteristics, grouping and intervention, assessment indicators, outcomes, and journals. The third researcher addressed differences regarding research design, looked over the article list and data extractions, and verified that there were no duplicate articles or patient records (Table 2). The quality of the literature was assessed using the MINORS scale [43]. There were 12 evaluation indications with a total score of 24 and a range of 0 to 2 points for each item. Zero denotes the absence of a report, one denotes a report with insufficient information, and two denotes a report with all available information. The last four indicators related to the additional items with the control group, while the first eight indicators related to the evaluation without the control group (Table 3). By using the Risk of Bias in Non-Randomized Intervention Studies (ROBINS-I) tool [44], two assessors separately evaluated the risk of bias for OMT outcomes. Finally, the variations were discussed. Table 4 and Table 5 and Figure 2 display evaluations of domain-specific and overall bias risk.

## 3. Results

### 3.1. Study Selection and Characteristics

The search technique returned a total of 1776 items. Following abstract and title screening, 146 papers were chosen for thorough assessment. Of these, 137 were eliminated because the article’s quality issues, systematic review, and meta-analysis were not suitable ending indicators. For the detailed analysis, nine were included [22,25,28,37,39,40,42,45]. Nine studies from 2013 to 2022, the majority of them in 2019, were included in our analysis. They were as follows: five prospective studies, one nonrandomized controlled trial, one controlled pre–post-controlled trial, one long-term follow-up study, and one comparative cohort study. Nine papers have a maximum score of 4.842 and the majority of their impact factors are from Region II. Children who were 3 years old or older were included to guarantee compliance with training. The intensity of the orofacial muscles, oral and maxillofacial function and morphology were the primary outcome markers used for the study. Image analysis, polysomnography, electromyography, and oral muscle function score evaluation were used as evaluation methods.

### 3.2. Types of Interventions

The study’s intervention programs were innovative in line with how each study was set up while ensuring that the overall goal remained the same. They were centered on the mechanistic elements of treatment and were designed utilizing a variety of practice models (Table 5). Passive training is more effective for youngsters in terms of compliance as compared to active training. There are advantages and downsides to having such a wide choice of intervention programs, including the advantage of better examining the best options and the disadvantage of not yet being able to locate an accepted OMT program for children. As a safety net for adherence issues, we also discovered that adding tools like daily punch cards, remote coaching, and smart aids can make the treatment effect more authentic. Parental participation is necessary for both active and passive actions to keep children adhering. This study found that specific training programs were more effective than no training programs, and the clinical scalability was higher.

### 3.3. Study Quality Assessment

The quality of the nine included studies was evaluated using the MINORS scale, which has a maximum score of 23, a minimum score of 14, and a mean score of 18.7 points. Three studies did not have a control group; therefore, the additional item was not applicable. Six studies included a control group, but the statistical analysis methods were not properly documented, which contributed to the quality disparities across the nine publications. None of the nine studies offered an estimated sample size.

### 3.4. Intervention Effect

Different assessment tools were used in nine papers to measure the craniomaxillofacial surface, the most common measurement tool being lateral cephalometrics evaluation, which uses imaging methods to reflect the facial morphology objectively by tracking changes in each marker point and marker line [46]. In a Chinese study, 12 iconic sites on the face were identified using photography techniques to track changes in facial morphology. The shape of the front and side profiles has been greatly improved. The significant difference was found in the proportion of Sn-Ls/Sn-Stms, Sn-Stms/Sn-Me, as well as in the angle of Gs-Sn-Pos, nasolabial angle, and mentolabial angle after OMT treatment [37]. Myofunctional therapy (MT) may be useful in the management of children with sleep-disordered breathing, according to Villa’s two prospective studies, each of which used a distinct set of assessment tools [22,28]. The Iowa Oral Performance Instrument (IOPI) objectively measures tongue and lip strength and endurance [47]. After two months of therapy, substantial differences between the MT and no-MT groups and the healthy children were found [28]. OMT resulted in a significant nasal congestion reduction, thereby reducing the proportion of positive Glazzel and Rosenthal tests, enabling patients to resume nasal breathing [22]. In addition, lip exercises allowed children to restore correct lip sealing. Cephalometric indicators of facial skeletal development defects, such as PNS-NPhp, PNS-AD2, and minRGA, have been improved.

Nine investigations were conducted, and only one demonstrated no improvement in orofacial dysfunction after OMT [25]; the other eight reported some improvement in facial morphology and function. The shortest time possible is two months, allowing for the observation of improvements in open-mouth breathing, facial muscular strength, and tongue function. Short-term therapies are only beneficial for improving muscle strength, and changes in facial appearance require long-term persistence. It could explain why the morphological improvements seen in studies with prolonged intervention or follow-up were more substantial. Habumugisha et al. [40] proposed that muscle function therapy plays a role in regulating lower facial height and fostering lateral maxillary arch development. However, Guilleminault et al. and Chuang et al. [25,39] found that the anterior facial height increased more vertically after OMT. The majority of research showed that breathing problems while sleeping improved after six months. After two months of OMT, muscle tension could be alleviated, according to two studies conducted in Villa in 2015 and 2017. However, a longer follow-up is frequently necessary to see improvements in the airway and craniofacial morphology. Following patients for up to 4 years, Guilleminault et al. [25] discovered that the upper airway was noticeably larger, and that the maxilla, mandible, and vertical facial height had all grown [37,38,42]. Others discovered an improvement in the skull and its appearance after at least half a year.

### 3.5. Bias Risk Assessment

Table 4 and Figure 2 show how the ROBINS-I instrument was utilized to assess bias risk. The vast majority of studies (55.6%) were considered to have a moderate risk of drift. This study included non-randomized controlled trials and confounding bias was assessed as moderate bias, mainly due to uncontrolled potential key confounders (i.e., age, gender, willingness, compliance, and other causes of craniomaxillofacial anomalies), which varied by study. The participant selection bias and missing data in the six trials were moderate, mainly due to the small sample size included in the study and the poor control of sample loss due to long-term follow-up. There are moderate deviations in outcome measurement in the eight studies, mainly because the outcome measurement indicators selected by each evaluator are inconsistent, and there may be certain measurement errors in objective examination results. Only in individual studies, to reduce the number of errors, did the same person conduct repeated measurements in different periods to take the average value. The relationship between the error of the measurement results and the intervention status is very small. Research participants’ understanding of the intervention measures will only have a small impact on the outcome measurement. In terms of implementation and compliance, there may be some bias in the results. Among the studies, only 33.3% posed a significant threat of bias. It is mainly reflected in confounding factors, selecting participants, and missing data. The proportion and reasons for participants missing in different intervention groups are slightly different, and analysis is unlikely to eliminate the risk of bias caused by data loss.

## 4. Discussion

The relationship between mouth breathing and orofacial development has been supported by a number of animal models developed in the 1980s [48]. Research on the effects of nasal breathing damage in children have also revealed that nasal respiratory impairment can affect quality of life [49] and face development. Currently, adenohysterectomy (AT) and orthodontic treatment are important components of the combined multidisciplinary treatment of OSAHS in children. AT surgery mainly relieves upper airway obstruction, and orthodontic treatment only alters the abnormal oropharyngeal structure, neither of which corrects abnormal neuromuscular function. One of the newest complementary therapies for sleep breathing disorders is OMT [23]. Myofunctional therapy purports to improve OSAHS by strengthening muscles, increasing the sensitivity and contraction of the orofacial and pharyngeal muscles, and maintaining the patency of the upper airway [50]. In addition, it can reposition the tongue [28], enhancing nasal breathing [25], and reduce submandibular fat to improve OSAHS [51]. The effectiveness of OMT in improving oxygen saturation and sleep quality, decreasing snoring, the apnea hypoventilation index, daytime sleepiness, and the recurrence of OSAHS in children after surgery has been shown in several studies over the past ten years [52]. It has also been shown to improve compliance with orthodontic appliances or CPAP [52]. Numerous comprehensive evaluations carried out by Camacho et al. [23,53] confirmed the effectiveness of OMT treatment on OSAHS in terms of hypoventilation index, snoring, and sleepiness.

The impact of OMT on children’s craniomaxillofacial region has not been the subject of any relevant research reviews, according to the study’s analysis of previously collected data. Based on this, this study applied the systematic review methodology to four electronic databases, using precise inclusion and exclusion criteria to conduct systematic searches. Due to the notable variety across the various forms of literature, qualitative analysis was only conducted on the selected material. It is not difficult to conclude from prior studies that OSAHS and orofacial muscle dysfunction are causally related [54,55]. In our investigation, we found that OMT improved facial morphology and supported the restoration of neuromuscular function. As a result, regardless of the role that facial neuromuscular function plays, OMT holds promise for both prevention and treatment. Guimares published data indicating that OMT has a limited role in adult OSAHS patients [56]. To promote normal airway development and guarantee that treatment has a lasting effect, Guilleminault stresses the significance of identifying and acting upon children with OSAHS as soon as possible [25]. Furthermore, children and their families must be involved to ensure training compliance, but compliance difficulties were not investigated in any of the nine studies, and in principle, poor compliance and insufficient intervention time can have an impact on the OMT treatment program. In actual practice, the OMT exercise routine and duration of therapy are frequently modified based on the needs and responses of each patient. There is controversy in the research regarding the best course of action for treating oral and facial dysfunction. The use of OMT has been the subject of an increasing number of studies in recent years, and multicenter studies are necessary to develop standardized OMT protocols that can be used either alone or in conjunction with other conventional and/or non-anatomical treatments to treat the illness.

National research on OMT has gradually increased over the past ten years, with a substantial amount of research conducted at the pathophysiological level [57], systematic reviews and meta-analyses [58,59], and intervention trials. Although various studies advocate for OMT as an adjunctive treatment for OSAHS, its effectiveness is controversial because only a few studies examined its effects with objective instruments. These disputes may be caused by differences in OMT training content, so in future practice, a unified and standardized training program needs to be established. From the current standpoint, there are numerous obstacles to promoting OMT. The first is how to formulate the scheme itself, which necessitates more randomized controlled trials and evidence summaries in clinical practice to formulate scientific schemes with an evidence-based approach; the second is how to ensure children’s compliance, which necessitates the establishment of a family participatory training model to ensure the training effect through incentives, supervision, feedback, remote guidance, and other means.

### Limitations

This review has several restrictions. Only one study from China was among the few which were accessible for inclusion. Data from China are now critically needed. To have a deeper understanding of OMT, it would be preferable to include as much research with a diverse geographic focus as possible. The majority of studies lacked more comprehensive patient data at the time of analyses, especially regarding clinical outcomes. This systematic analysis is based on non-RCTs, which lack randomization and are, therefore, less reliable than RCTs. These investigations are more prone to statistical problems as a result. Finally, only a few research investigations have highlighted the importance of training compliance.

## 5. Conclusions

According to the findings of this study, OMT can improve the strength, shape, and function of oral and facial muscles, as well as have a positive impact on children’s growth and development. With a better understanding of these effects, the next step in the research is to develop a systematic OMT treatment plan for prevention and treatment to improve breathing in children by optimizing oral and facial development. Future pediatric research will focus on identifying and treating pediatric OSA through a collaborative and interactive approach between otolaryngologists, orthodontists, pulmonary allergists, sleep physicians, endocrinologists, orofacial muscle function therapists, and speech therapists [60].

## Figures and Tables

**Figure 1 children-10-00670-f001:**
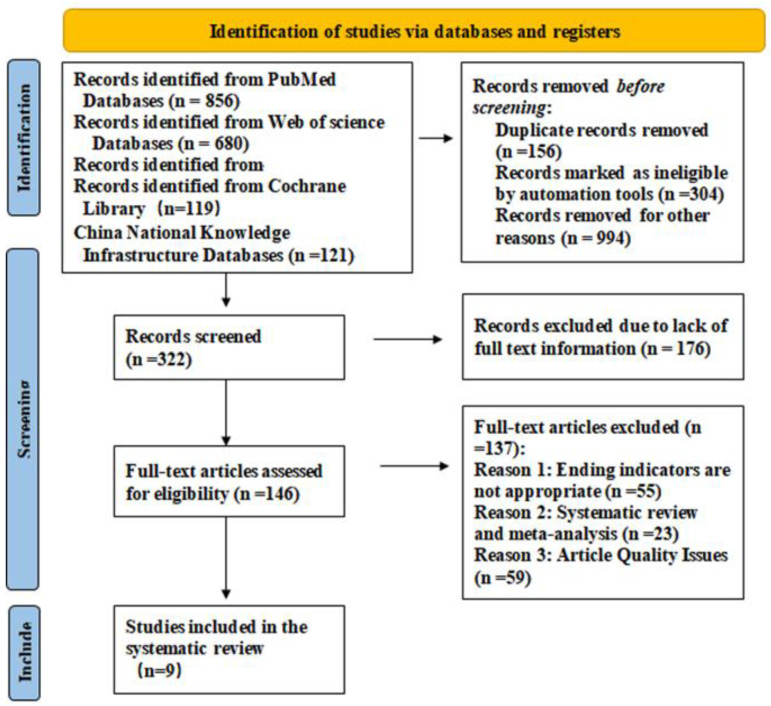
PRISMA 2020 flow diagram.

**Figure 2 children-10-00670-f002:**
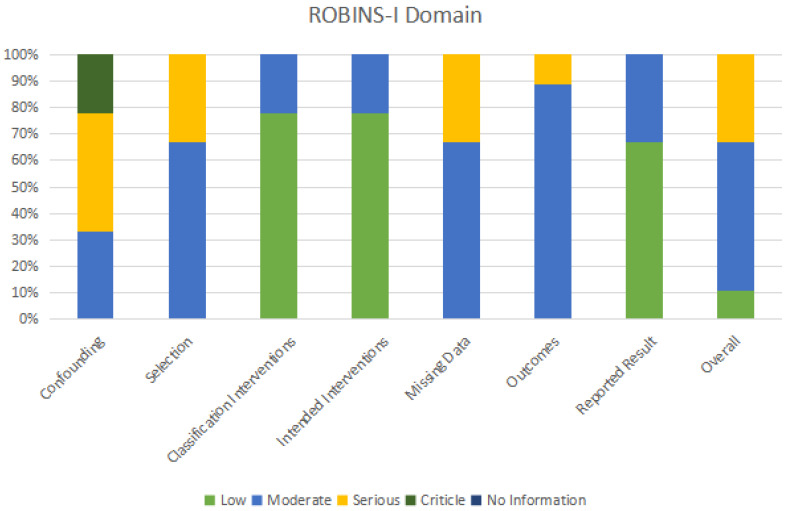
Utilizing the ROBINS-I instrument to assess bias risk.

**Table 1 children-10-00670-t001:** Electronic database search strategies.

Electronic Database	Search Strategy
PubMed (searched up to 1 September 2022)	All Fields: respiratory muscle therapy OR oropharyngeal exercises OR speech therapy OR breathing exercises OR wind musical instruments OR orofacial myofunctional therapy OR Myofunctional therapy OR Muscle function therapy OR Oropharyngeal movements AND Craniofacial Language: English and Chinese
The Web of Science (searched up to 1 September 2022)	Keywords: respiratory muscle therapy OR, oropharyngeal exercises OR speech therapy OR breathing exercises OR wind musical instruments OR orofacial myofunctional therapy OR Myofunctional therapy OR Muscle function therapy OR Oropharyngeal movements AND Craniofacial
Cochrane Library	orofacial myofunctional therapy OR Myofunctional therapy OR Oropharyngeal movements AND OSA AND Child
China National Knowledge Infrastructure (CNKI)	Keywords: orofacial myofunctional therapy OR Oropharyngeal movements OR speech therapy AND child

**Table 2 children-10-00670-t002:** Characteristics of the included studies.

Study Citation (Year)	Type of Study Design	Sample Size and Age	Intervention	Assessments	Primary Findings	Journals
Shan, 2021 [37]	Self-controlled before and after study	10 children aged 4–7 years, 7 boys and 3 girls.	The study group performed 4 sets of exercises daily and required parental participation and supervision for 6 months: (1) labial training; (2) breathing training; (3) tongue position training; and (4) swallowing training	Take photos before and after orofacial myofunctional therapy: twelve representative mark points	The shape of the front and side profiles has been greatly improved. the significant difference was found in the proportion of Sn-Ls/Sn-Stms, Sn-Stms/Sn-Me, as well as in the angle of Gs-Sn-Pos, nasolabial angle, mentolabial angle after OMT treatment.	*Shanghai Journal of Stomatology*
Guilleminault, 2013 [25]	Retrospective analysis	24 cases of children aged 3–6 years with OSAHS	Eleven cases in the study group were given OMT and followed up at 22–50 months after treatment; the control group had a blank intervention and was followed up at 20–34 months.	Sleep-related questionnaire, PSG, orofacial myofunctional evaluation with scores	Children with SDB have abnormal upper airway muscle contractions while they slumber. All participants were given a score of abnormal orofacial muscle tone while awake at the conclusion of the assessment by re-educators.	*SLEEP MED*
Huang, 2019 [38]	Prospective study	Total 121 pediatric OSA. Fifty-four children were received MFT.	MFT group: MFT (total 20 min/d) for 0.5 years, For 0.5 years, the PMFT group received mouth applications with tongue beads while they slept. Control group: received no medical attention.	PSG, lateral cephalometric films evaluating bone structure development	MFT showed improvement in PNS-NPhp and PNS-AD2 measurements	*Sleep Med Clin*
Chuang, 2019 [39]	Comparative cohort study	57 children (44 males and 13 girls, mean age 7.86 ± 3.09 years old)	PMFT group (Oral appliance for 1 year) Control group: Blank control	PSG, Quality of life survey (the Chinese version of OSA-18). Cephalometric radiography	The upper airways’ dimensions (PNS-AD2, minRGA, OPha-Ophp) considerably grew. Mandibular and maxillary development (Ar-A) (Ar-Gn and Co-Gn). vertical development of facial height (S-Go and N-Me)	*Sleep & breathing = Schlaf & Atmung*
Habumugisha, 2022 [40]	non-randomized concurrent controlled trial	224 kids (aged 6 to 10); 114 boys and 110 girls.	MB-N group (mouth breathers with no medication, n = 70); NB group (nasal breathers with no treatment, n = 79); MB-M group (mouth breathers with myofunc-tional treatment, n = 75): With a therapy of 13.0 ± 1.1 months.	nasopharyngeal X-ray, rhinoscopy, and flexible nasopharyngoscopy	U1-NA, L1-NB angles and U1-NA, L1-NB linear measures declined in the MB-M group, while overjet, over-bite, and C-C linear lengths rose.	*BMC PEDIATR*
Huang, 2019 [41]	a long term follow-up study	110 children aged 4 to 16 years with PSG diagnosis of OSA	Group MFT: a total of 20 minutes per day for a year. Oral application using a tongue bead for a year in the PMFT cohort.	PSG and lateral cephalometrics evaluating bone structure development at baseline, 6 and 12 months.	The airway width at the level of the nasopharynx (min-RGA, PNS-AD2) was significantly enhanced by the oral device. Improvements in the PNS-NPhp and PNS-AD2 measurements are Particularly significant. With PMFT, compliance is much easier.	*SLEEP MED*
Hwang, 2022 [42]	retrospective study	50 boys and 13 girls, totaling 63 patients with OSAHS, were between the ages of 4 and 16 years.	For more than six months, each patient received nightly PMFT (OA) therapy.	Lateral Cephalometric Radiography	28 patients reacted favorably to the therapy. In respondents, there were larger SNBa, smaller LGo an-gle, and shorter SN. A tongue-beaded OA can indicate a positive outcome for pediatric OSAHS with a smaller LGo Angle and smaller SN.	*Children*
Villa, 2015 [22]	prospective and randomized study	30 OSAHS children over 4 years old	group 1 (n = 14): oropharyngeal exercises + nasal washing control group (n = 13): nasal washing	Glatzel and Rosenthal tests, polysomnography and clinical evaluation	A decrease in oral breathing, satisfactory Glatzel and Rosenthal tests, and a noticeable enhancement in labial seal and lip tone.	*Sleep & breathing = Schlaf & Atmung*
Villa, 2017 [28]	prospective, case–control study	54 kids with SDB (mean age 7.1 ± 2.5 years, 29 boys)	MT group (n = 36): MT plus nasal washing no-MT group (n = 18): nasal washing	Myofunctional evaluation tests, the Iowa Oral Performance Instrument (IOPI), and nocturnal pulse oximetry	Boosted tongue strength, peak tongue pressure and persistence, restored normal tongue resting posture, and decreased oral breathing and lip hypotonia.	*Sleep & breathing = Schlaf & Atmung*

Abbreviations: polysomnography (PSG); myofunctional therapy (MFT); passive myofunctional therapy (PMFT); oral appliance (OA); Cranial base angulation in midsagittal plane (SNBa); lower gonial angle (L Go Angle); Anterior cranial base length, from sella to nasion (SN); Sleep disordered breathing (SDB).

**Table 3 children-10-00670-t003:** Application of MINORS in Literature Quality Evaluation.

Entry	Shan, 2021 [37]	Guilleminault, 2013 [25]	Huan, 2019 [38]	Chuang, 2019 [39]	Habumugisha, 2022 [40]	Huang, 2019 [41]	Hwang, 2022 [42]	Villa, 2015 [22]	Villa, 2017 [28]
Q1	2	2	2	2	2	2	2	2	2
Q2	2	2	2	2	2	2	2	2	2
Q3	2	2	2	2	2	2	2	2	2
Q4	2	2	2	2	2	2	2	2	2
Q5	2	2	2	2	2	2	2	2	2
Q6	2	2	2	2	2	2	2	2	2
Q7	2	2	2	2	2	2	2	2	2
Q8	0	0	0	0	2	0	0	0	0
Q9	Not applicable	Not applicable	2	2	2	2	Not applicable	2	2
Q10	Not applicable	Not applicable	2	2	2	2	Not applicable	2	2
Q11	Not applicable	Not applicable	2	2	2	2	Not applicable	2	2
Q12	Not applicable	Not applicable	1	1	1	1	Not applicable	0	0
Total score	14	14	21	21	23	21	14	20	20

Notes: Q1: The purpose of the study is clearly stated; Q2: Consistency of inclusion of patients; Q3: Collection of expected data; Q4: Outcome indicators can appropriately reflect the research purpose; Q5: Objectivity of evaluation of outcome indicators; Q6: Whether the follow-up time is sufficient; Q7: Lost interview rate is lower than 5%; Q8: Whether the sample size is estimated; Used to evaluate additional entries with comparison groups Q9: Whether the control group is properly selected; Q10: Whether the control group is synchronized; Q11: Whether the baseline between groups is comparable; Q12: Whether the statistical analysis is appropriate.

**Table 4 children-10-00670-t004:** Process for evaluating the ROBINS-I instrument.

Rank	Evaluation Category	Shan, 2021 [37]	Guilleminault, 2013 [25]	Huang, 2019 [38]	Chuang, 2019 [39]	Habumugisha, 2022 [40]	Huang, 2019 [41]	Hwang, 2022 [42]	Villa, 2015 [22]	Villa, 2017 [28]
Bias due to confounding	1.1	4	1	1	1	2	2	4	1	2
	1.2	9	0	9	1	2	1	9	1	1
	1.3	9	2	9	1	1	1	9	1	1
	1.4	0	2	2	9	9	9	3	9	9
	1.5	2	2	4	9	9	9	9	9	9
	1.6	0	3	2	9	9	9	4	9	9
	1.7	3	3	3	3	2	4	3	3	3
	1.8	9	9	9	9	2	9	9	9	9
	Risk of bias judgement (direction)	4(1)	2(1)	3(1)	3(1)	2(1)	3(1)	4(1)	3(1)	2(1)
Bias in selection of participants into the study	2.1	4	3	3	4	4	4	4	4	4
	2.2	9	9	9	9	9	9	9	9	9
	2.3	9	9	9	9	9	9	9	9	9
	2.4	2	1	2	1	1	1	2	1	1
	2.5	9	9	9	9	9	9	9	9	9
	Risk of bias judgement (direction)	3(4)	3(1)	2(1)	2(1)	2(1)	2(1)	3(1)	2(1)	2(1)
Bias in classification of interventions	3.1	1	1	1	1	1	1	1	1	1
	3.2	0	1	1	1	1	1	1	1	1
	3.3	2	1	4	3	3	3	4	3	3
	Risk of bias judgement (direction)	2(2)	2(2)	1	1	1	1	1	1	1
Bias due to deviations from intended interventions	4.1	9	9	9	4	4	4	9	4	4
	4.2	9	9	9	9	9	9	9	9	9
	4.3	2	0	1	9	9	9	1	9	9
	4.4	1	1	1	9	9	9	2	9	9
	4.5	2	1	1	9	9	9	2	9	9
	4.6	9	1	9	9	9	9	9	9	9
	Risk of bias judgement (direction)	2(2)	1	1	1	1	1	2(2)	1	1
Bias due to missing data	5.1	1	1	1	1	1	1	1	1	1
	5.2	3	4	1	4	1	1	1	4	4
	5.3	3	4	1	0	2	2	1	4	4
	5.4	9	9	2	9	0	2	2	9	9
	5.5	9	9	4	9	2	2	4	9	9
	Risk of bias judgement (direction)	2(2)	3(1)	2(1)	2(1)	2(2)	2(1)	2(1)	3(1)	3(1)
Bias in measurement of outcomes	6.1	3	2	3	4	4	4	3	4	4
	6.2	1	1	1	1	1	1	1	1	1
	6.3	4	4	4	1	1	1	4	1	1
	6.4	4	4	4	1	3	3	4	3	3
	Risk of bias judgement (direction)	2(4)	2(2)	2(1)	3(2)	2(2)	2(1)	2(2)	2(2)	2(2)
Bias in selection of the reported result	7.1	1	1	1	4	4	4	4	4	4
	7.2	4	4	4	4	4	4	4	4	4
	7.3	4	4	4	1	1	1	1	1	1
	Risk of bias judgement (direction)	2(4)	2(1)	2(1)	1	1	1	1	1	1

**Table 5 children-10-00670-t005:** OMT intervention plan.

Study Citation (Year)	Intervention Program
Shan, 2021 [37]	(1) Four types: Lip muscle training; Breathing training; Tongue position training; Swallowing training (2) Lip and breathing training for the first 1 to 3 months and tongue and swallowing training for the last 4 to 6 months. Parents were asked to record their children’s training on video for 15 to 20 min a day.
Guilleminault, 2013 [25]	Orthodontic treatment and myofunctional re-education are done concurrently. There was no mention of a training program.
Huang, 2019 [38]	MFT (20 min/day total) for 0.5 years (1) Soft palate movements: continuously (isometric exercises) or sporadically pronounce oral vowel sounds (isotonic exercises). (2) Tongue exercises include moving the tongue along the lateral and superior surfaces of the teeth, pressing the tongue’s complete length against the hard and soft palates, and squeezing the tongue against the floor of the mouth. (3) Facial exercises: lateral jaw movements, suction motions, intraoral finger pressure against the buccinator muscles, and tension and relaxation of the orbicularis oris. (4) Stomatognathic duties include alternate-side chewing and swallowing with the teeth clamped together, the tongue in the palate, and no perioral muscle contraction.
Chuang, 2019 [39]	Study participants were required to install instruments and beads (passive MFT) with tongues at night. Parents recorded the night wear of children in the treatment group for one year in the sleep log. We will schedule three months of recall for each participant in order to confirm the status and installation status of the oral device and the side effects and discomfort when installing the device. Fix or adjust the oral device if necessary.
Habumugisha, 2022 [40]	(1) Lip sealing exercises using a lip trainer with a tension of 250 g, spanning 10 mi, three times each day, are performed by a professional dental nurse. One hundred times per day, rotate the tongue (the tip of the tongue beats fiercely in the palate); 15 times a day, sprinkle gum on the upper mandible as part of your gum training; Swallowing instruction (push 15 mL of water on the tip of their tongue against the hard palate, shut their lips and swallow), 15 times a day. (MRC-I stage, MRC-II stage, training for lip sealing with a lip trainee, training for tongue flipping, training for chewing gum, and practice for swallowing). (2) An orthodontic trainee was demonstrated to the kids. Every day and night, children must wear the trainer for two hours. After two weeks, the initial examination must be performed; thereafter, it must be done every four weeks. The therapy period lasts 1–1.5 years.
Huang, 2019 [41]	(1) MFT consists of movements that are isotonic and isometric and target the oropharynx and oral structures. Parents were requested to supervise children for at least 20 min of exercise each day after initial training with a specialist. (2) Oral appliance: The patient had to use the device every night, put it in his mouth before bed, and move the beads in his tongue before sleeping (i.e., passive MFT). For a full year, parents maintain sleep diaries to track their kids’ nighttime activities. A monthly equipment inspection is performed by the dentist.
Hwang, 2022 [42]	Over the course of more than six months, all patients received OA therapy each night. For the tongue tip to roll, the tongue bead is mounted on the bottom end of the frame. To open the airway, the mandible of the wearer is positioned forward and downward, and the mouth is positioned forward. The OA was put on by the patients before bed, and they were told to roll the bead with their tongues while they slept. Every three months, each patient was given a reminder appointment to examine the fit and condition of their oral device and any side effects or discomfort they may have experienced while using it.
Villa, 2015 [22]	(1) Three types: mouth posture exercises, labial seal and lip toning exercises, and nasal breathing rehabilitation. (2) Youngsters must exercise at least three times per day, 10 to 20 times each, at home. Twice a month, the MT team visits with the therapist for muscle function.
Villa, 2017 [28]	Same as Villa, 2015 [22]

## Data Availability

The datasets used to conduct the present study can be found in the PubMed repository at URL (accessed on 27 March 2023).

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
