# Peer review of "The Effects of Orofacial Myofunctional Therapy on Children with OSAHS’s Craniomaxillofacial Growth: A Systematic Review"

_children, 2023, doi:10.3390/children10040670_

Round 1

Reviewer 1 Report

Line 9: remove two points after abstract 

Line 10: start with capital letter

Line 17: metheds 

In the introduction you have to explain the different phenotypes 

Tan HL, Kaditis AG. Phenotypic variance in pediatric obstructive sleep apnea. Pediatr Pulmonol. 2021 Jun;56(6):1754-1762. doi: 10.1002/ppul.25309. Epub 2021 Feb 16. PMID: 33543838.

and the relationship with bruxism

Segù M, Pollis M, Santagostini A, Meola F, Manfredini D. Correlation between Parental-Reported Tooth Grinding and Sleep Disorders: Investigation in a Cohort of 741 Consecutive Children. Pain Res Manag. 2020 Jul 30;2020:3408928. doi: 10.1155/2020/3408928. PMID: 32802220; PMCID: PMC7415113.

Line 38 remove two points

Line 39 remove two points

Author Response

Dear Professor

Thank you for allowing our manuscript ID children-2234763 entitled " The effects of orofacial myofunctional therapy on children with OSAHS's craniomaxillofacial growth: a systematic review (children-2234763)" to be revised and submitted to Children (ISSN 2227-9067). We greatly appreciate the time and effort of editors and referees toward evaluating and revising our research work.

Please refer to the attached for a point-to-point response to the comments that are raised. Accordingly, the manuscript has been revised with additional data and text changes marked in blue. We hope that the improved version would be now suitable for publication.

All authors have read the whole manuscript with approval for publication, and this article was not submitted to elsewhere at the same time.

Thank you for your consideration. Should you have further concerns, please do not hesitate to contact me.

With best wishes,

Sincerely,

Jian-Rong Zhou, M.D.

Chongqing Medical University

1 Youyi Road, Yuzhong District, Chongqing, 400016, China.

E-mail: 202028@cqmu.edu.cn

Tel: 86-023-63555767; Fax: 86-023-63555767  

Responses to the Reviewers’ Comments

Reviewers' comments:

We have made extensive modifications to our manuscript and supplemented extra data to make our results convincing. The reviewer comments are laid out below in italicized font and specific concerns have been numbered. Our response is given in normal font and changes/additions to the manuscript are given in blue text.

Reviewer #1:     

  1. Line 9: remove two points after abstract

Answer: We have revised it according to the journal specifications.

  1. Line 10: start with capital letter

Answer: We were really sorry for our careless mistakes. Thank you for your reminder. we have corrected the “orofacial” into “Orofacial”.

  1. Line 17: metheds

Answer: We have re-written this part according to the Reviewer’s suggestion. In the part of methodology, in order to highlight the process of systematic review, we deleted the description part of the database and added the principles of PICO to make the editor more clear about our research purpose. For the detailed description of PICO, please see lines 99-103 of the text.

  1. In the introduction you have to explain the different phenotypes and the relationship with bruxism

Answer: Many thanks to the editor for pointing out the shortcomings of our research background. Your suggestions and guidance will help us re-interpret the background and significance of this research. As two types of sleep disorders, OSAHS and sleep bruxism may lead to craniomaxillofacial abnormalities. In the introduction, we rearranged the logical thinking, introduced OSAHS and sleep bruxism from sleep disorders, summarized their status and the mechanism that may cause craniofacial abnormalities, and introduced the principle and efficacy of OMT. See lines 32-45 for details.

  1. Line 38 remove two points

Answer: Many thanks to the editor for his professional opinion, we have removed the two points in line 38.

  1. Line 39 remove two points

Answer: In order to better meet the requirements of periodical typesetting, we have deleted the subtitle and presented the introduction as a complete story line.

We appreciate for Editors/Reviewers’ warm work earnestly, and hope the correction will meet with approval. Once again, thank you very much for your comments and suggestions.

Reviewer 2 Report

Dear Authors,

Orofacial myofunctional therapy (OMT) is one of the therapeutic methods for neuromuscular re-education, and has been considered as one of the auxiliary methods for obstructive sleep apnea hypopnea syndrome (OSAHS) and orthodontic treatment. This systematic review examines the literature on the craniomaxillofacial effects of OMT in children with OSAHS  to lay the foundation for further optimization of orofacial muscle function treatment protocols and to promote the use of OMT in children.

The study is of scientific interest and in line with the aims of the journal. 

However, there are some issues that should be addressed.

First of all, the template layout was totally wrong. Please modify.

Abstract

  • Abstract: The abstract should be a total of about 200 words maximum. The abstract should be a single paragraph and should follow the style of structured abstracts, BUT WITHOUT HEADINGS: 1) Background: Place the question addressed in a broad context and highlight the purpose of the study; 2) Methods: Describe briefly the main methods or treatments applied. Include any relevant preregistration numbers, and species and strains of any animals used. 3) Results: Summarize the article's main findings; and 4) Conclusion: Indicate the main conclusions or interpretations. The abstract should be an objective representation of the article: it must not contain results which are not presented and substantiated in the main text and should not exaggerate the main conclusions. Please follow these instructions. (https://www.mdpi.com/journal/children/instructions)
  • Few of the studies reviewed reported the effects of OMT 12 on the musculature.” Please revise English language.
  • Line 14. Please report OSAHS acronym.
  • Please report the PICO in the method.

Introduction

  • Introduction section should be a single paragraph. (https://www.mdpi.com/journal/children/instructions)
  • Line 41. “high prevalence and serious long term 41 complications.”, Please add reference. Moreover, please better report the definition and report recent epidemiological data.
  • Line 44-49. This sentence is too long and not clear.
  • English language should be revised by a mother-tongue.
  • Line 68-73. Please report recent literature on this topic, report the gap in the scientific literature to justify your paper, and report the rationale of the study.

 Materials and Methods

·      Did you register the systematic review on PROSPERO or others?

·      Lines 102-103. These outcomes were not clear.

·      Report references in all Tables. In Table 1, Remove IF. Add a column with Journals.

Discussion and References were well written.

Author Response

Dear Professor

Thank you for allowing our manuscript ID children-2234763 entitled " The effects of orofacial myofunctional therapy on children with OSAHS's craniomaxillofacial growth: a systematic review (children-2234763)" to be revised and submitted to Children (ISSN 2227-9067). We greatly appreciate the time and effort of editors and referees toward evaluating and revising our research work.

Please refer to the attached for a point-to-point response to the comments that are raised. Accordingly, the manuscript has been revised with additional data and text changes marked in blue. We hope that the improved version would be now suitable for publication.

All authors have read the whole manuscript with approval for publication, and this article was not submitted to elsewhere at the same time.

Thank you for your consideration. Should you have further concerns, please do not hesitate to contact me.

With best wishes,

Sincerely,

Jian-Rong Zhou, M.D.

Chongqing Medical University

1 Youyi Road, Yuzhong District, Chongqing, 400016, China.

E-mail: 202028@cqmu.edu.cn

Tel: 86-023-63555767; Fax: 86-023-63555767  

Responses to the Reviewers’ Comments

Reviewers' comments:

Reviewer #2:

Orofacial myofunctional therapy (OMT) is one of the therapeutic methods for neuromuscular re-education, and has been considered as one of the auxiliary methods for obstructive sleep apnea hypopnea syndrome (OSAHS) and orthodontic treatment. This systematic review examines the literature on the craniomaxillofacial effects of OMT in children with OSAHS to lay the foundation for further optimization of orofacial muscle function treatment protocols and to promote the use of OMT in children. The study is of scientific interest and in line with the aims of the journal. However, there are some issues that should be addressed.

Answer: we sincerely thank the editor and all reviewers for their valuable feedback that we have used to improve the quality of our manuscript and help us better understand our story. The reviewer comments are laid out below in italicized font and specific concerns have been numbered. Our response is given in normal font and changes/additions to the manuscript are given in blue text.

We feel great thanks for your professional review work on our article. As you are concerned, several problems need to be addressed. According to your nice suggestions, we have made extensive corrections to our previous draft, the detailed corrections are listed below.

  1. Abstract

1.1 The template layout was totally wrong. Please modify.

Answer: When the manuscript was in the revision stage, we typeset the manuscript according to the journal guidelines, and we used the Children's Microsoft Word template file. The specific changes to the layout are as follows: (1) Change to a structured summary format with a word limit of 200 (2) Change the introduction to a whole paragraph (3) add author contributions and Data Availability Statement.  

1.2 The abstract should be a total of about 200 words maximum. The abstract should be a single paragraph and should follow the style of structured abstracts, BUT WITHOUT HEADINGS: 1) Background: Place the question addressed in a broad context and highlight the purpose of the study; 2) Methods: Describe briefly the main methods or treatments applied. Include any relevant preregistration numbers, and species and strains of any animals used. 3) Results: Summarize the article's main findings; and 4) Conclusion: Indicate the main conclusions or interpretations. The abstract should be an objective representation of the article: it must not contain results which are not presented and substantiated in the main text and should not exaggerate the main conclusions. Please follow these instructions. (https://www.mdpi.com/journal/children/instructions)

Answer: We have abridged the abstract as appropriate. The lack of systematic review articles on the effects of OMT on the craniomaxillofacial region was highlighted in the background section as a way to support this study. PICO principles were added in the Methodology section to screen the literature. A brief description of the main outcome indicators is presented in the results section. The discussion section provides a theoretical summary of the study and development prospect of OMT. For details, please refer to lines 11-27 of the revised draft.

1.3 Few of the studies reviewed reported the effects of OMT 12 on the musculature.”

Please revise English language.

Answer: Change “Few of the studies reviewed reported the effects of OMT on the musculature” to “There is a dearth of comprehensive analysis of OMT's effects on muscle morphology and function” in line 14.

1.4 Line 14. Please report OSAHS acronym.

Answer: As for the referee's concern, the full descriptions of the abbreviations like obstructive sleep apnea hypopnea syndrome (OSAHS) have been supplemented in the revised manuscript. For details, please refer to line 13 of the revised manuscript.

1.5 Please report the PICO in the method.

Answer: PICO (patients, intervention, comparison, outcomes) principles have been added in line 17, for details refer to the methods section of the text (lines 99-103).

  1. Introduction

2.1 Introduction section should be a single paragraph.

(https://www.mdpi.com/journal/children/instructions)

Answer: We have reorganized the logical structure of the introduction to present it as a cascading paragraph. The rationale for the treatment of OMT and its efficacy is presented in the context of the association between Sleep-disordered breathing and the craniomaxillofacial area.

2.2 Line 41. “high prevalence and serious long term 41 complications.”, Please add

reference. Moreover, please better report the definition and report recent

epidemiological data.

Answer: As suggested by the reviewer, we have added more references to support this idea. In the introduction, to make the research background more complete and clear, we sorted out various phenotypes of sleep-disordered breathing, extracted two categories that may have an impact on craniofacial and maxillofacial as the entry point of the story, described the prevalence rate and possible mechanism of action, introduced the principle of OMT as an adjuvant therapy, and described its application effect (lines 32-45).

The additional references:

  1. Xu, Z.-G.; Xu, J.-J.; Chen, Y.-C.; Hu, J.; Wu, Y.; Xue, Y. Aberrant Cerebral Blood Flow in Tinnitus Patients with Migraine: A Perfusion Functional MRI Study. J. Headache Pain 2021, 22, 61, doi:10.1186/s10194-021-01280-0.
  2. Zinchuk, A.V.; Gentry, M.J.; Concato, J.; Yaggi, H.K. Phenotypes in Obstructive Sleep Apnea: A Definition, Examples and Evolution of Approaches. Sleep Med Rev 2017, 35, 113–123, doi:10.1016/j.smrv.2016.10.002.
  3. Tan, H.-L.; Kaditis, A.G. Phenotypic Variance in Pediatric Obstructive Sleep Apnea. Pediatr Pulmonol 2021, 56, 1754–1762, doi:10.1002/ppul.25309.
  4. Paglia, L. Respiratory Sleep Disorders in Children and Role of the Paediatric Dentist. Eur J Paediatr Dent 2019, 20, 5, doi:10.23804/ejpd.2019.20.01.01.
  5. Sateia, M.J. International Classification of Sleep Disorders-Third Edition: Highlights and Modifications. Chest 2014, 146, 1387–1394, doi:10.1378/chest.14-0970.
  6. Singh, S.; Kaur, H.; Singh, S.; Khawaja, I. Parasomnias: A Comprehensive Review. Cureus 2018, 10, e3807, doi:10.7759/cureus.3807.
  7. Manfredini, D.; Restrepo, C.; Diaz-Serrano, K.; Winocur, E.; Lobbezoo, F. Prevalence of Sleep Bruxism in Children: A Systematic Review of the Literature. J Oral Rehabil 2013, 40, 631–642, doi:10.1111/joor.12069.
  8. Waters, K.A.; Suresh, S.; Nixon, G.M. Sleep Disorders in Children. Med J Aust 2013, 199, S31-35, doi:10.5694/mja13.10621.
  9. Stark, T.R.; Pozo-Alonso, M.; Daniels, R.; Camacho, M. Pediatric Considerations for Dental Sleep Medicine. Sleep Med Clin 2018, 13, 531–548, doi:10.1016/j.jsmc.2018.08.002.

The definition of sleep disorders includes various types, among which OSAHS and sleep bruxism may have an impact on the craniofacial region. The incidence of OSAHS is shown in line 41, and the prevalence of sleep bruxism is shown in line 43.

2.3 Line 44-49. This sentence is too long and not clear.

Answer: We are very sorry, in order to make the sentence more concise and understandable, we have revised as follows: The relationship between sleep problems and craniofacial features is debatable, most studies agree that breathing patterns influence the function and morphology of craniomaxillofacial muscles, and directly affect the stability and collapse of the upper airway during sleep.

2.4 English language should be revised by a mother-tongue.

Answer: Thanks for your suggestion. We tried our best to improve the manuscript and made some changes to the manuscript. These changes will not influence the content and framework of the paper. And here we did not list the changes but marked in blue in the revised paper. We appreciate for Editors/Reviewers’ warm work earnestly and hope that the correction will meet with approval.

2.5 Line 68-73. Please report recent literature on this topic, report the gap in the

scientific literature to justify your paper, and report the rationale of the study.

Answer: We sincerely appreciate the valuable comments. We have checked the literature carefully and added more references on Efficacy of OMT and Shortcomings of existing studies into the INTRODUCTION part in the revised manuscript (lines 50-61; 69-85)

The additional references:

  1. Yuen, H.M.; Chan, K.C.; Chu, W.C.W.; Chan, J.W.Y.; Wing, Y.K.; Li, A.M.; Au, C.T. Craniofacial Phenotyping by Photogrammetry in Chinese Prepubertal Children with Obstructive Sleep Apnea. Sleep 2022, zsac289, doi:10.1093/sleep/zsac289.
  2. Motta, L.J.; Martins, M.D.; Fernandes, K.P.S.; Mesquita-Ferrari, R.A.; Biasotto-Gonzalez, D.A.; Bussadori, S.K. Craniocervical Posture and Bruxism in Children. Physiother. Res. Int. 2011, 16, 57–61, doi:10.1002/pri.478.
  3. Guilleminault, C.; Huang, Y.S.; Monteyrol, P.J.; Sato, R.; Quo, S.; Lin, C.H. Critical Role of Myofascial Reeducation in Pediatric Sleep-Disordered Breathing. Sleep Medicine 2013, 14, 518–525, doi:10.1016/j.sleep.2013.01.013.
  4. Galeotti, A.; Festa, P.; Pavone, M.; De Vincentiis, G.C. Effects of Simultaneous Palatal Expansion and Mandibular Advancement in a Child Suffering from OSA. Acta Otorhinolaryngol Ital 2016, 36, 328–332, doi:10.14639/0392-100X-548.
  5. Hamoda, M.M.; Almeida, F.R.; Pliska, B.T. Long-Term Side Effects of Sleep Apnea Treatment with Oral Appliances: Nature, Magnitude and Predictors of Long-Term Changes. Sleep Med 2019, 56, 184–191, doi:10.1016/j.sleep.2018.12.012.
  6. Fagundes, N.C.F.; Flores-Mir, C. Pediatric Obstructive Sleep Apnea-Dental Professionals Can Play a Crucial Role. Pediatr Pulmonol 2022, 57, 1860–1868, doi:10.1002/ppul.25291. 3.                                                           3.Materials and Methods

3.1 Did you register the systematic review on PROSPERO or others?

Answer: We think this is an excellent suggestion and we have applied to PROSPERO, but it has not been approved yet. If there is news we will be the first to inform you.

3.2 Lines 102-103. These outcomes were not clear.

Answer: Our primary outcome indicators include three areas: (1)Cephalometric indicators: SNA, SNB, ANB, PP-MP , SN-MP , SN-PP , SN-OP , OP-MP , FMA, N-Me, SN-Gn, SNGoGn, GoGn, ArGoMe, ArGo, N-ANS, ANS-Me, S-Go, MP-H, 1-NA, 1. NA, 1. NB, NB, SPAS, PAS, C3-H, overbite, and overjet et al. (2)Muscle function assessment:the Iowa Oral Performance Instrument (IOPI) and orofacial myofunctional evaluation with scores(OMES). (3)Sleep breathing assessment:Apnea hypopnea index (AHI) in polysomnography (PSG); disease-specific quality of life for children with obstructive sleep apnea 18 items survey (OSA-18). See lines 102-110 and Table 2 for more details.

3.3 Report references in all Tables. In Table 1, Remove IF. Add a column

with Journals.

Answer: We have presented the references in the table. We have removed the entry IF and added the journal name according to the Reviewer’s suggestion.

  1. Discussion and References

4.1 were well written

Answer: We thank the reviewer for reading our paper carefully and giving the above positive comments.

We appreciate for Editors/Reviewers’ warm work earnestly, and hope the correction will meet with approval. Once again, thank you very much for your comments and suggestions.

Reviewer 3 Report

Dear authors thanks for your very pertinent work. After some alterations, I consider this work an important starting point for future studies in the field. 

You can find the comments in the file. If you agree, consider the suggestions like opportunities to clarify and improve your work.

Kind Regards

Author Response

Dear Professor

Thank you for allowing our manuscript ID children-2234763 entitled " The effects of orofacial myofunctional therapy on children with OSAHS's craniomaxillofacial growth: a systematic review (children-2234763)" to be revised and submitted to Children (ISSN 2227-9067). We greatly appreciate the time and effort of editors and referees toward evaluating and revising our research work.

Please refer to the attached for a point-to-point response to the comments that are raised. Accordingly, the manuscript has been revised with additional data and text changes marked in blue. We hope that the improved version would be now suitable for publication.

All authors have read the whole manuscript with approval for publication, and this article was not submitted to elsewhere at the same time.

Thank you for your consideration. Should you have further concerns, please do not hesitate to contact me.

With best wishes,

Sincerely,

Jian-Rong Zhou, M.D.

Chongqing Medical University

1 Youyi Road, Yuzhong District, Chongqing, 400016, China.

E-mail: 202028@cqmu.edu.cn

Tel: 86-023-63555767; Fax: 86-023-63555767  

Responses to the Reviewers’ Comments

Reviewers' comments:

Reviewer #3:

1.Dear authors thanks for your very pertinent work. After some alterations, I consider this work an important starting point for future studies in the field. 

Answer: We are honored to receive your recognition, which encourages us to continuously improve our study design and explore appropriate training programs in clinical practice. The reviewer comments are laid out below in italicized font and specific concerns have been numbered. Our response is given in normal font and changes/additions to the manuscript are given in blue text.

2.You can find the comments in the file. If you agree, consider the suggestions like opportunities to clarify and improve your work.

2.1 "as" in capital "AS".

Answer: Many thanks to the editor for pointing out our spelling mistakes. In order to make the content more concise in the new revised manuscript, we have deleted this part of content.

2.2 Orofacial in capital.

Answer: According to the writing requirements of the introduction, we condensed it into a complete paragraph and deleted the subheadings. Thanks to the editor's reminder, we have double-checked the spelling problems in the revised manuscript.

2.3 "improve" or contribute to control and decrease the consequences of OSHA?

Answer: Many thanks to the editor for his careful reading and keen insight, which is indeed not clearly described. There is still some controversy about the efficacy of OMT. OMT can achieve short-term control of OSAHS symptoms, and long-term OMT is needed to continuously improve symptoms. Currently, OMT alone or in combination has achieved positive results in controlling sleep breathing symptoms, improving craniomaxillofacial cephalometric indices, enhancing muscle thickness, and boosting muscle activity. (lines 70-73) 

2.4 All the tables are results, so in my oppinion should just be showned and refered in the results part of the manuscript.

Answer: As the editor said, most of the data in Table 2 can guide the writing of the results, so we added the key data in the table to the results.

2.5 and what was diferent in this study? Techniques? Characteristics of the sample? What explain this "out-side"?

Answer: Thanks to the editors for their guiding comments on the results section, which prompted us to rethink why improvements were observed in these 8 studies. We re-examined these 8 studies with such questions in mind. Finally, we found that the common point of the eight studies was that the training compliance was good, which was reflected in parental participation and passive OMT, so its improvement effect was significant, while the other study did not guarantee compliance well. It just highlights our research conclusion again: compliance is the key to ensuring the efficacy of OMT.  

We appreciate for Editors/Reviewers’ warm work earnestly, and hope the correction will meet with approval. Once again, thank you very much for your comments and suggestions.

Reviewer 4 Report

Dear Authors, 

This is a Systematic Review is attractive and it is considered of scientific and even more clinical interest.

However, some suggestions should be made. 

The English language must be revised by a professional in order to improve grammar and vocabulary through the whole study. I recommend to avoid long sentences: short sentences are easier to understand.

The Abstract Section is well organized, and it clearly describes the research. 

The Introduction is well structured and explains the study rationale, starting from previous Systematic Reviews.

But I suggest to improve it these articles will be useful, add some lines about muscles activity, iperactivity and possible Orofacial pain; and differences with other oral appliances doi: 10.3390/medicina59020410 ;

The Material and Methods section is adequate and well organized, but punctuation and spaces between words should be reviewed.

The PICO questions is not clear

In the results and in the materials Add the risk of bias using Rob 2 of the included studies

Finally, I suggest a Minor Revision for this Research. 

Best regards

Author Response

Dear Professor

Thank you for allowing our manuscript ID children-2234763 entitled " The effects of orofacial myofunctional therapy on children with OSAHS's craniomaxillofacial growth: a systematic review (children-2234763)" to be revised and submitted to Children (ISSN 2227-9067). We greatly appreciate the time and effort of editors and referees toward evaluating and revising our research work.

Please refer to the attached for a point-to-point response to the comments that are raised. Accordingly, the manuscript has been revised with additional data and text changes marked in blue. We hope that the improved version would be now suitable for publication.

All authors have read the whole manuscript with approval for publication, and this article was not submitted to elsewhere at the same time.

Thank you for your consideration. Should you have further concerns, please do not hesitate to contact me.

With best wishes,

Sincerely,

Jian-Rong Zhou, M.D.

Chongqing Medical University

1 Youyi Road, Yuzhong District, Chongqing, 400016, China.

E-mail: 202028@cqmu.edu.cn

Tel: 86-023-63555767; Fax: 86-023-63555767  

Responses to the Reviewers’ Comments

Reviewers' comments:

Reviewer #4:

  1. This is a Systematic Review is attractive and it is considered of scientific and even more clinical interest. However, some suggestions should be made. 

Answer: Thank you for your affirmation of the significance of our research, which has doubled our confidence. We have repeatedly considered your question and revised the research.

  1. The English language must be revised by a professional in order to improve grammar and vocabulary through the whole study. I recommend to avoid long sentences: short sentences are easier to understand.

Answer: Thanks for your suggestion. We tried our best to improve the manuscript and made some changes to the manuscript. These changes will not influence the content and framework of the paper. And here we did not list the changes but marked in blue in the revised paper. We appreciate for Editors/Reviewers’ warm work earnestly and hope that the correction will meet with approval.

  1. The Abstract Section is well organized, and it clearly describes the research. 

Answer: Thank you for your affirmation of the summary. We have revised the abstract according to the opinions of several other editors, mainly by deleting the unimportant description, refining the main research process, and limiting the number of words to 200 according to the writing method of the structured abstract.

  1. The Introduction is well structured and explains the study rationale, starting from previous Systematic Reviews. But I suggest to improve it these articles will be useful, add some lines about muscles activity, iperactivity and possible Orofacial pain; and differences with other oral appliances doi: 10.3390/medicina59020410

Answer: Thanks to the editor's interpretation and combing of the introduction, we also recognize the lack of overview on oral-facial muscle dysfunction and oral orthotics. The references provided by the editor have great inspiration for our research ideas. (lines 45-50; 61-64)

Additional References:

  1. Gomes, M.F.; Giannasi, L.C.; Fillietaz-Bacigalupo, E.; de Mancilha, G.P.; de Carvalho Silva, G.R.; Soviero, L.D.; da Silva, G.Y.S.; Nazario, L. de M.; Dutra, M.T.D.S.; Silvestre, P.R.; et al. Evaluation of the Masticatory Biomechanical Function in Down Syndrome and Its Influence on Sleep Disorders, Body Adiposity and Salivary Parameters. J Oral Rehabil 2020, 47, 1007–1022, doi:10.1111/joor.13023.
  2. Kang, J.-H.; Kim, H.J. Potential Role of Obstructive Sleep Apnea on Pain Sensitization and Jaw Function in Temporomandibular Disorder Patients. J Korean Med Sci 2022, 37, e307, doi:10.3346/jkms.2022.37.e307.

  1. The Material and Methods section is adequate and well organized, but punctuation and spaces between words should be reviewed.

Answer: Thank you for your detailed comments. We have realized the English format problem in the first draft and marked it with blue font in the revised manuscript.

  1. The PICO questions is not clear

Answer: We have realized that the PICO principle is not standardized when writing the article, and several other editors have also raised the same question, so we have described the meaning of PICO in detail in the new manuscript. (lines 99-103)

  1. In the results and in the materials Add the risk of bias using Rob 2 of the included studies

Answer: Thank you very much for pointing out the shortcomings of our research. We used the Risk of Bias in Non-Randomized Intervention Studies (ROBINS-I) tool for assessment because the articles included in our research are non-randomized controlled trials. The particular evaluation procedure is detailed in Table 5, and the evaluation results are shown in Figure 2.

  1. Finally, I suggest a Minor Revision for this Research. 

Answer: We appreciate for Editors/Reviewers’ warm work earnestly, and hope the correction will meet with approval. We have revised many parts of the manuscript, added supporting data and improved sentence structure. Once again, thank you very much for your comments and suggestions.

Round 2

Reviewer 2 Report

Authors modified the text according to the suggestions.

I found this work impactful and it fits well with in the scope of this journal.

However, the paper was not registered on PROSPERO.

Author Response

Dear Professor

Thank you for allowing our manuscript ID children-2234763 entitled " The effects of orofacial myofunctional therapy on children with OSAHS's craniomaxillofacial growth: a systematic review (children-2234763)" to be revised and submitted to Children (ISSN 2227-9067). We greatly appreciate the time and effort of editors and referees toward evaluating and revising our research work.

Please refer to the attached for a point-to-point response to the comments that are raised. Accordingly, the manuscript has been revised with additional data and text changes marked in blue. We hope that the improved version would be now suitable for publication.

All authors have read the whole manuscript with approval for publication, and this article was not submitted to elsewhere at the same time.

Thank you for your consideration. Should you have further concerns, please do not hesitate to contact me.

With best wishes,

Sincerely,

Jian-Rong Zhou, M.D.

Chongqing Medical University

1 Youyi Road, Yuzhong District, Chongqing, 400016, China.

E-mail: 202028@cqmu.edu.cn

Tel: 86-023-63555767; Fax: 86-023-63555767  

Responses to the Reviewers’ Comments

Reviewers' comments (round 2):

Authors modified the text according to the suggestions. I found this work impactful and it fits well with in the scope of this journal.

Reviewer #2 (round 2):     

  1. However, the paper was not registered on PROSPERO.

 We had to pay for our negligence first. When the editors first informed us that a registration statement was required, we registered right away with PROSPERO. During the revision of the manuscript, the editor also sends email reminders, and we also pay attention to the updated information on the email and official website at any time. Until today, we found that the research proposal was not submitted but saved, which is a serious mistake. We would like to express our gratitude to the editor for continually reminding us. We are ready to bear any repercussions resulting from our carelessness, but we would appreciate it if you could give us some time to wait for the registration, and we will know the outcome in about ten days.  

 We have conducted a registration search on PROSPERO. The effects of OMT on OSAHS in adults and children have been examined similarly in recent literature. They do not pay enough attention to the function and morphology of the craniomaxillofacial area, instead concentrating mostly on sleep breathing, and snoring. Based on this, this research examines the effects of OMT on the morphology and outcome indicators of the craniomaxillofacial region. To sum up, this study has some research value that can demonstrate OMT's effectiveness in a variety of fields and advance clinical practice research.

We appreciate for Editors/Reviewers’ warm work earnestly, and hope the correction will meet with approval. Once again, thank you very much for your comments and suggestions.
